# Potential Activity of Abrantes Pollen Extract: Biochemical and Cellular Model Studies

**DOI:** 10.3390/foods10112804

**Published:** 2021-11-15

**Authors:** Ana C. Gonçalves, Radhia Aitfella Lahlou, Gilberto Alves, Cristina Garcia-Viguera, Diego A. Moreno, Luís R. Silva

**Affiliations:** 1CICS–UBI–Health Sciences Research Centre, Department of Medical Science, University of Beira Interior, 6201-506 Covilhã, Portugal; anacarolinagoncalves@sapo.pt (A.C.G.); gilberto@fcsaude.ubi.pt (G.A.); 2Chemistry Department, University of Beira Interior, 6201-001 Covilhã, Portugal; ra.aitfella@ubi.pt; 3Fiber Materials and Environmental Technologies (FibEnTech), University of Beira Interior, 6201-001 Covilhã, Portugal; 4Laboratory of Valorisation and Conservation of Biological Resources, Biology Department, Faculty of Sciences, University M’Hamed Bougara, Boumerdes 35000, Algeria; 5Phytochemistry and Healthy Foods Laboratory, Food Science and Technology Department, CEBAS-CSIC, Campus Universitario de Espinardo, 30100 Espinardo, Murcia, Spain; cgviguera@cebas.csic.es; 6CPIRN-UDI-IPG–Research Unit for Inland Development, Center for Potential and Innovation of Natural Resources, Polythecnic of Guarda, 6300-559 Guarda, Portugal

**Keywords:** bee pollen, phenolic characterization, biological potential, functional food, antioxidant effects

## Abstract

The aim of this study was to determine the grain composition and (poly)phenolic profile of pollen from Abrantes (Portugal), as well as its antioxidative and antidiabetic properties, and abilities to protect human erythrocytes against induced hemoglobin oxidation, lipid peroxidation, and hemolysis. The phytochemical profile of the Abrantes’ bee pollen revealed twenty phenolic compounds, identified by high-performance liquid chromatography with electrospray ionization mass spectrometry coupled with photodiode array detection. Among them, quercetin derivatives were the most abundant. Concerning the biological potential, the pollen extract showed notable capacity for 2,2-diphenyl-1-picrylhydrazyl, nitric oxide, and superoxide radicals, as well as for inhibition of α-glucosidase action, and protection of human erythrocytes against oxidative damage. Non-cytotoxic effects regarding the NHDF normal cell line, human adenocarcinoma Caco-2, and human liver HepG2 cells were observed. The results obtained contributed to further research on modes of action related to oxidative damage and metabolic health problems, to generate deeper knowledge of potential health-promoting effects to develop novel pharmaceutical drugs, nutraceuticals, and dietary supplements.

## 1. Introduction

Nowadays, there is an interest in more balanced, functional, and healthier food. In this context, natural plants and their derivatives have received a lot of attention owing to their health-promoting properties with little or no adverse effects [1]. Bee pollen, a hive-derived product, is one of these products that is gaining more attention [2]. In fact, it has been used since early times in complementary and folk medicine to attenuate several pathological conditions, such as anemia, allergies, colds, colitis, enteritis, flu, and ulcers [3]. It consists of flower pollen collected from various selected botanical sources by the honeybee *Apis mellifera,* with the purpose of feeding larvae in their early stages of development [4,5].

Bee pollen is constituted of about 200 biologically active substances, collected from plants and transported to the hive in the form of pollen loads [2]. The general composition of pollens roughly consists of polysaccharides (50%), simple sugars (mainly fructose and glucose at percentages varying from 4 to 10%), fats and lipids (1–20%), proteins (6–28%), and amino acids (6%), along with several secondary plant metabolites such as phenolic compounds (flavonoids and phenolic acids), which are the main contributors of their color and bitter taste, carotenoids, and terpenes [2]. Additionally, pollen constituents can be useful chemotaxonomic markers, as well as quality indicators [5]. 

Regarding phenolics, pollen has higher concentrations of flavonols, namely quercetin derivatives, with values ranging from 24.0 (quercetin) to 956 µg/g (quercetin 3-*O*-rutinoside) [6]. Furthermore, it is also high in several essential elements, including calcium, copper, magnesium, iron, and zinc [7]. In fact, bee pollen has been widely acclaimed owing to its nutritional and therapeutic properties, and is already commercialized as a functional food and recognized as a medicinal product by the German Federal Board of Health [2,8]. In general, it can be consumed fresh, dried, or incorporated in beverages, tonics, bars, granules, tablets, and nutraceuticals [2]. Its notable antioxidant, anti-inflammatory, antimicrobial, antifungal, antiviral, anti-mutagenic, and anti-allergic potential are mainly attributed to the presence of phenolics, which are considered the main active compounds in this foodstuff [3]. 

Therefore, given the current interest in pollen and the existence of few studies on this natural product, the aim of this work was to determine pollen grain composition and the phenolic profile of multifloral bee pollen from Abrantes (Portugal). This city is located in the province of Ribatejo and belongs to Santarém district. Considering its favorable edaphoclimatic conditions and flora variety (e.g., *Castanea sativa, Cytisus, Echium, Quercus* and *Rubus* plants) which confer to this regional honey unique characteristics, including a dark color and intense flavor, while not being too sweet, it is not surprising that honey production is increasing in this region [9,10]. Furthermore, its biological potential was also evaluated, namely the capacity to scavenge 2,2-diphenyl-1-picrylhydrazyl (DPPH^●^), nitric oxide (^●^NO), and superoxide (O_2_^●−^) radicals and to inhibit the *α*-glucosidase enzyme. Additionally, its protective effects on different cellular models were also studied: (1) human erythrocytes, studying hemoglobin oxidation, lipid peroxidation, and hemolysis induced by peroxyl species (ROO^●^); (2) human colon carcinoma cells (Caco-2); and (3) liver carcinoma cells (HepG2). Erythrocytes were selected not only for being the most abundant cells in vertebrates, but also for their crucial role in immune reactions and respiratory gas exchange, which in turn, makes them highly susceptible targets to oxidation and inflammatory conditions. On the other hand, Caco-2 and HepG2 cancer cells were chosen as they present many morphological characteristics of intestinal epithelium and liver parenchymal cells, respectively. For comparative purposes, the effects on primary normal human dermal fibroblasts (NHDF) were studied.

## 2. Materials and Methods

### 2.1. Chemical Reagents

All chemicals used were of analytical grade.

4-Nitrophenyl-alpha-D-glucopyranoside (pNPG), N-(1-naphthyl)ethylenediamine dihydrochloride, sodium nitroprusside dihydrate (SNP), and sulfanilamide were acquired from Alfa Aesar (Karlsruhe, Germany). Acetonitrile and methanol were purchased from Fisher Chemical (Leicestershire, UK). The remaining ones were from Sigma-Aldrich (St. Louis, MO, USA). Water was deionized using a Milli-Q water purification system (Millipore Ibérica, S.A.U., Madrid). Caco-2, HepG2 and NHDF cell lines were from American Type Culture Collection (Manassas, VA, USA).

### 2.2. Pollen Samples

The bee pollen sample (REF: M08AG006), composed of at least 500 pollen grains, was directly provided by a local company (Colmeicentro) in Alferrarede, Abrantes region, in 2017. The sample was stored in the dark under desiccating conditions to avoid alterations until use.

### 2.3. Pollen Analysis

The botanical origin of the sample was determined according to the melissopalynological method described by Louveaux, Maurizio, & Vorwohl [11]. Briefly, 10 g of the sample was diluted with bidistilled water (50 mL) and centrifuged at 3900× *g* for 10 min in order to separate the pollen grains. Then, the obtained sediment was again dissolved in water and centrifuged for 5 min. Finally, two aliquots of the sediment were examined microscopically at 45×, using a bright-field microscope (Olympus, Tokyo). Each aliquot was composed of a minimum of 800 pollen grains. The results were expressed as percentages.

### 2.4. Pollen Extract Preparation

The extract was prepared according to Moita et al. [3]. Briefly, 0.2 g of bee pollen were thoroughly mixed in 1 mL of ethanol:water (70:30, *v/v*), ultrasonicated for 1 h and centrifuged at 2900× *g* during 10 min at room temperature. Then, the supernatant was evaporated under reduced pressure to complete dryness at 40 °C. The resulting concentrate residue was stored at −20 °C, and protected from light until use. The obtained extraction yield from the starting dry material was 64.5 ± 0.16%. The extractions were performed in triplicate.

### 2.5. Identification of Phenolic Compounds via HPLC-DAD-ESI/MS^n^

The identification of phenolic compounds from bee pollen was performed according to Gonçalves et al. [12]. They were tentatively identified based on their ultraviolet-visible and mass spectra features, elution order, and retention times as compared to authentic standards analyzed the under same conditions (Table 1), and also with data available in the literature [12,13,14,15,16]. Injections were performed in triplicate.

### 2.6. Quantification of Phenolic Compounds via HPLC-DAD

Concerning phenolics quantification, 20 µL extract was injected on an LC model Agilent 1260 system (Agilent, Santa Clara, CA, USA), according to Gonçalves et al. [5]. Spectral data from all peaks were accumulated in a range varying from 200 to 600 nm and chromatograms were recorded at 320 and 350 nm for hydroxycinnamic acids and flavonols, respectively. Compounds were identified by comparing their retention times and ultraviolet-visible spectra with those of authentic standards. Peak purity was checked by software contrast facilities. Caffeoyl di-hexose and caffeoyl hexose were quantified as caffeic acid, and coumaroyl hexose as *ρ*-coumaric acid. Quercetin 7-glucoside-3-*O*-rutinoside, quercetin derivatives 1 and 2, quercetin hexoside and quercetin acethylrhamnoside were quantified as quercetin. Kaempferol di-hexoside, kaempferol 3-*O*-rutinoside-*O*-heoxide, kaempferol acethylhexoside, myrcetin rhamno-hexoside, and myrcetin derivative were quantified as kaempferol 3-*O*-rutinoside. On the other hand, isorhamnetin 3-*O*-rutinoside and isorhamnetin acetyl hexoside were quantified as quercetin. Analyses were carried out in triplicate.

### 2.7. Antioxidant Capacity Experiments

The capacity of bee pollen hydroethanolic extracts to scavenge free radicals, namely DPPH^●^, ^●^NO and O_2_^●−^, was performed spectrophotometrically through in vitro microassays using 96-well plates. Microplate reader Bio-Rad Xmark spectrophotometer (Bio-Rad Laboratories, Hercules, CA, USA) was used to measure the absorbances. The results were expressed as 25% and 50% maximal inhibitory concentration (IC_25_ and IC_50_, respectively) values (µg/mL). Each experiment was performed in triplicate, and seven different concentrations were tested.

#### 2.7.1. 1,1-Diphenyl-2-Picrylhydrazyl Radical (DPPH^●^) Assay

The capability of bee pollen hydroethanolic extract to quench DPPH^●^ was performed according to Gonçalves et al. [1]. Each well contained 25 µL extract dissolved in methanol and 200 µL methanolic DPPH (150 mM). Control was composed of replacing the sample with water, while blank was composed of water, or diluted extract, and methanol. The absorbances were taken at 515 nm. Ascorbic acid was used as positive control.

#### 2.7.2. Nitric Oxide Radical (^●^NO) Assay

The effects of bee pollen extracts in capturing ^●^NO were based on the work of Silva & Teixeira [17]. Briefly, each well was composed of 100 µL each extract dissolved in potassium phosphate buffer (100 mM, pH 7.4) and 100 µL of sodium nitroprusside dihydrate (20 mM). Blanks and controls contained phosphate buffer and sodium nitroprusside dihydrate. The plates were incubated at room temperature for one hour, under light. Then, 100 µL Griess reagent (1% sulfanilamide and 0.1% naphthylethylenediamine in 2% H_3_PO_4_) was added to each well and incubated for 10 min in a dark (blanks received 100 µL H_3_PO_4_). After this time, the absorbance was read at 560 nm. Ascorbic acid was used as positive control.

#### 2.7.3. Superoxide Radical (O_2_^●−^) Assay

The possibility of pollen extracts being able to reduce O_2_^●−^ generated by the NADH/phenazine methosulfate system was done according to a previous method [1]. Each well contained 50 µL each diluted extract dissolved in potassium phosphate buffer (19 mM, pH 7.4), 50 µL *β*-nicotinamide adenine dinucleotide (NADH), 150 µL nitrotetrazolium blue chloride (NBT), and 50 µL phenazine methosulfate (PMS). The control was composed of phosphate buffer instead of diluted extract. After the addition of PMS, each plate was immediately collocated in the spectrophotometer and absorbance readings were collected every 10 seg over 2 min at 562 nm. Ascorbic acid was used as a positive control.

### 2.8. α-Glucosidase Inhibitory Activity

The inhibition of *α*-glucosidase was determined based on a previous method [17]. Seven different concentrations were done. Each well contained 150 µL potassium phosphate buffer (19 mM, pH 7.4), 50 µL each concentration dissolved in buffer, and 100 µL 4-nitrophenyl-*α*-D-glucopyranoside (PNP-G). Control was only composed of phosphate buffer and PNP-G. The reaction was initiated by the addition of 25 µL *α*-glucosidase in each well, followed by a period of incubation of 10 min of incubation at 37 °C. Finally, and after that time, the absorbance of 4-nitrophenol released from PNP-G was read at 405 nm. Acarbose was used as positive control. Three experiments were performed in triplicate.

### 2.9. Biological Activity on Cellular Models

#### 2.9.1. Isolation of Human Erythrocytes

Venous human blood was collected from randomized patients of Cova da Beira Hospital Centre (Covilhã, Portugal) by antecubital venipuncture into K_3_EDTA vacuum tubes, in accordance with the protocol approved by the Health Ethical Commission of Cova da Beira Hospital Centre (no. 25/2018) without the need for written informed consent of the patients. Therefore, erythrocytes from blood collected in the early morning already processed by the hospital and without any need for more studies. Then, they were isolated according to Gonçalves et al. [1]. Briefly, 4 mL of a collected sample was transferred to sterile conic tubes, mixed with 6 mL PBS at pH 7.4 and centrifuged at 1500× *g* for 5 min at 4 °C. This procedure was repeated three more times, and the resulting supernatant from each centrifugation was always discarded. The number of cells (cells/mL) and viability (always above 98%) were obtained by the Trypan blue exclusion method. The suspensions of isolated erythrocytes were kept on ice until use.

#### 2.9.2. Peroxyl Radical (ROO^●^)-Induced Oxidative Damage in Human Erythrocytes

For evaluating the protective effects of pollen hydroethanolic extract against ROO**^●^** species in human erythrocytes, six different concentrations were prepared and dissolved in PBS. Five experiments were performed in duplicate in 48-well microplates.

#### 2.9.3. Inhibition of Hemoglobin Oxidation

The capacity of pollen extracts to prevent methemoglobin oxidation induced by 2,2′-azobis(2-amidinopropane) dihydrochloride (AAPH) was based on a previous work [1]. Six dilutions of the extract were prepared in PBS. The reaction was composed of 100 µL sample solution and 200 µL erythrocytes solution (1250 × 10^6^ cells/mL, final density). Controls and blanks were performed by replacing the sample with 100 µL PBS. The mixture was then incubated in a water bath for 30 min at 37 °C, under slow agitation. After the incubation, 200 µL AAPH (50 mM, final concentration) prepared in PBS were added to the media (except in the blank), followed by another incubation using the same conditions described above for 4 h. After that time, the entire volume of the reaction mixture was transferred to conic Eppendorf tubes and centrifuged at 1500× *g* for 5 min at 4 °C. Finally, 300 µL supernatant was placed in a 96-well plate and the absorbance was measured at 630 nm. Quercetin was used as a positive control.

#### 2.9.4. Inhibition of Lipid Peroxidation

The ability of the extract to avoid lipid peroxidation was indirectly monitored through the formation of thiobarbituric-acid-reactive substances (TBARS) [18]. Six different concentrations were prepared in PBS. First, 100 µL of each concentration was mixed with 200 µL of erythrocyte suspension (500 × 10^6^ cells/mL, final density) and incubated at 37 °C for 30 min, under slow agitation. After this period, 200 µL *tert*-BHP (0.2 mM, final concentration) was added to the media, and each well was further incubated under the same conditions for 30 min. Finally, the entire content was transferred to conic Eppendorfs, mixed with 250 µL trichloroacetic acid 28% (*w/v*) to incite protein precipitation, and centrifuged at 16,000× *g* for 10 min at 18 °C. Then, the supernatant was collected in conical test tubes (with screw caps) and 125 µL thiobarbituric acid 1% (*w/v*) was added to generate TBARS. The resulting mixture was heated for 15 min at 100 °C in a water bath. Finally, test tubes were cooled in ice and the absorbance was taken at 532 nm. Quercetin was used as a positive control.

#### 2.9.5. Inhibition of Hemolysis

The capacity of pollen extracts to avoid erythrocyte lysis induced by AAPH was evaluated by monitoring the release of hemoglobin to the media after membrane disruption, according to a previous work [19]. Briefly, six different dilutions were prepared in PBS. The mixture consisted of a 200 µL suspension of human erythrocytes (1775 × 10^6^ cells/mL, final density) and 100 µL of each concentration. Controls and blanks were composed of PBS. In a first step, the reaction mixture was incubated at 37 °C in a water bath for 30 min, under slow agitation. After this period, 200 µL AAPH (17 mM, final concentration) was added to the mixture (except in the blank), followed by another period of incubation for 3 h, using the aforementioned conditions. Lastly, the entire volume was then transferred to a conic Eppendorf and centrifuged at 1500× *g*, for 5 min at 4 °C. After that, 300 µL supernatant was transferred to 96-well plates and the absorbance was read at 540 nm. Quercetin was used as a positive control.

### 2.10. Cancer Cell Models

#### 2.10.1. Cell Culture Conditions and Treatments

To understand the possible cytotoxicity effects, Caco-2, HepG2, and NHDF cell lines were cultured in 75 cm^2^ culture flasks and incubated at 37 °C in a humidified atmosphere of 5% CO_2_. Caco-2 and HepG2 cells were cultured as a monolayer in DMEM containing 10% FBS, 1% penicillin/streptomycin, while NHDF cells were cultured in RPMI 1640 medium supplemented with FBS (10%), l-glutamine (2 mM), 4-(2-hydroxyethyl)-1-piperazineethanesulfonic acid (10 mM), sodium pyruvate (1 mM), and penicillin/streptomycin (1%). The medium was changed every 2 days. After a few passages, cells were washed twice with PBS and detached by gentle trypsinization. Then, viable cells were counted and suitably diluted in the adequate complete culture medium (25,000 cells/mL for Caco-2 cells, and 10,000 cells/mL for HepG2 and NHDF cells) [1,20]. After counting the cells, 200 µL of prepared cellular suspension was seeded in 96-well plates and incubated for one day before carrying out the viability assays. After 24 h of incubation, 200 µL of different concentrations of pollen extracts were dissolved in medium containing 0.5% (*v/v*) DMSO (the final concentration of DMSO did not affect cellular viability (data not shown)) and plates were again incubated for a further 24 h. At the end of this period, the metabolic activity of cells was evaluated concerning their ability to reduce yellow tetrazolium 3-(4,5-dimethylthiazol-2-yl)-2,5-diphenyltetrazolium bromide (MTT) to a blue formazan product and to ensure membrane integrity. Cell culture conditions and procedures were consistent through all assays. All studies were conducted when cells were in the logarithmic growth phase.

#### 2.10.2. MTT Assay

After 24 h of cell treatment with different concentrations of pollen extracts, the medium was rejected, and each well was washed with PBS. Then, cultures were incubated with 200 µL MTT solution (dissolved in appropriate serum-free medium) for 4 h at 37 °C. After this period, the medium containing MTT was removed, and the purple crystals of formazan were dissolved in DMSO. The absorbance of the different solutions was read at 570 nm. Cell proliferation values were expressed as percentages from the relative absorbance measured in the treated wells versus control wells [21]. A total of six independent experiments per extract were performed. Untreated cells were used as control.

#### 2.10.3. Membrane Integrity Assay

In order to evaluate the release of the stable cytosolic enzyme lactate dehydrogenase (LDH) into the medium, after 24 h of cells incubation with the extracts, 50 µL culture medium was placed to a 96-well plate. Then, 200 µL NADH (252.84 mM) and 25 µL pyruvate (14.99 mM) were added to each well, according to Gonçalves et al. [1]. Both pyruvate and NADH solutions were prepared in PBS (pH 7.4). The LDH released was determined at 340 nm due to NADH oxidation during the conversion of pyruvate to lactate, in a kinetic mode. A decrease in absorbance is directly related to the quantity of LDH released by the cells in the culture environment. A total of six independent experiments were performed. Untreated cells were used as control.

### 2.11. Statistical Analysis

Statistical comparison was determined using one-way ANOVA and the means were classified by Tukey’s test at a 95% level of significance. Differences were considered significant for *p* < 0.05. To determine the contribution of the total phenols, on their antioxidant activity, Pearson’s correlation coefficients were calculated. All analyses were carried out using Graph Pad Prism Version 6.01 (GraphPad Software, Inc., San Diego, CA, USA).

## 3. Results and Discussion

### 3.1. Pollen Composition

As expected, grains from different plant species were identified, which allows the pollen sample to be classified as heterofloral [8] (Figure 1). Particularly, thirteen different grain types were distinguished in the studied sample. Among them, *Echium plantagineum, Cistus ladanifer* and *Quercus* species were the predominant ones, with relative abundance percentages of 26, 23, and 19%, respectively. Other pollen types detected include *Rubus* spp., *Eriobotrya* spp., *Raphanus raphanistrum*, *Crepis capillaris*, *Anthyllis* spp., *Trifolium* spp., *Cytisus* spp., *Lavandula stoechas, Calendula arvensis, Salix* spp. And *Lotus criticus.* The obtained results are in line with the wide plant diversity observed in the Abrantes region. In fact, this area has a mixed forest of *Echium, Castanea sativa* and *Quercus* spp., Leguminosae scrubs (*Cytisus* spp.), and *Rubus* plants, which often colonize abandoned agricultural fields [9]. Additionally, the presence of these grain types would contribute to the dark blue color exhibited by honeys from this region [10]. These pollen grains have also been identified in other Portuguese samples, but at different proportions, depending on the region [2,5,9]. Particularly, honeys from Trás-os-Montes and Serra de Aire e Candeeiros are also richer in *Echium plantagineum* pollen grains [9,22], while honeys from Castelo Branco are mainly composed of *Lavandula* and *Echium* types (93.75 and 81.25%, respectively) [10]. On the other hand, Caramulo and Luso regions showed higher amounts of *Eucalyptus* spp. [5,23], while *C. sativa* is the most abundant grain type in Bragança samples [24]. Regarding other countries, *Rubus*, *C. sativa,* and *Cytisus* species are the predominant pollen grains in Spanish honeys [25], while Brazilian ones mainly consist of *Arecaceae, Asteraceae* spp., and *Brassicaceae* grains [8]. In addition to those already mentioned, it is important to note pollen composition is not only strongly influenced by floral species, but also by growth conditions and cultivation techniques [8]. On the other hand, its quality mainly depends on the harvesting, cleanliness, dryness, and storage procedures applied by beekeepers to increase shelf-life [2].

### 3.2. Phenolic Characterization

In this study, twenty phenolic compounds were tentatively identified by the interpretation of their fragmentation patterns obtained from mass spectra and by comparison with literature. Particularly, three hydroxycinnamic acids and seventeen flavonols were detected, as listed in Table 1. The total amount was 4.17 µg/g of dry weight (fw). As far as we know, this is the first study concerning the phenolic profile of bee-collected pollen from the Abrantes region (Portugal). Phenolic presence contributes to bee products appearance and functional properties, boosting antioxidant, anti-mutagenic, antimicrobial, and anti-inflammatory effects [5,26]. Their levels are strongly influenced by floral sources, geographical origin, bee species, and environmental factors, such as temperature, humidity, soil type, and storage time [7].

#### 3.2.1. Hydroxycinnamic Acids

Three hydroxycinnamic acids were found (Table 1, peaks 1–3). Regarding caffeoyl-derivatives, two signal peaks were detected (peaks 1 and 3), accounting for less than 1.34% of total phenolic compounds identified. Both showed characteristic fragments at *m/z* 179, 342 and 135, which indicates the presence of a caffeic acid and the consequent decarboxylation of this moiety [12,15]. As far as we know, this is the first time that these compounds have been reported in bee-pollen samples. Even so, the presence of caffeic acid was already reported in pollen (1.05 µg/mL of dw) [14] and honey (ca. 2.42 µg/mL of dw) [5,9,25].

Additionally, one coumaroyl hexose was found (peak 2), exhibiting a molecular ion at *m/z* 325 and a fragment pattern at *m/z* 163, which is characteristic of coumaroyl compounds [12,16]. Other researchers already reported the presence of coumaroyl acids in pollen and honey samples at residual amounts [9,14,16].

In accordance with the literature, it is possible to state that hydroxycinnamic acids, even in vestigial amounts, increase the antioxidant potential of bee-pollen products, essentially due to their CH = CH-COOH group [27].

#### 3.2.2. Flavonol Glycosides

Seventeen different flavonol glycosides were analyzed, including seven quercetin glycosides (peaks 4, 7, 9, 13, 15, 16 and 20), five kaempferols (peaks 5, 8, 12, 18 and 19), two myricetins (peaks 6 and 10), and three glycosides of isorhamnetin (peaks 11, 14 and 17).

Seven isomers of quercetin were identified. The majority displayed molecular ions at *m*/*z* 463 or 609 and fragment ion at *m/z* 300/301, which corresponds to quercetin aglycon losses [12,13,14,15,16]. These compounds were the most abundant phenolic compound found in the studied sample, which is in line with previous reports [8,13,14,16]. Trace amounts of quercetin derivatives were also found in honey samples [13].

Additionally, five kaempferol derivatives were found. Most of them showed molecular ions and fragment ions at *m/z* 447 and 285, respectively, which are characteristics of this compound, as previously reported [12,14,15,16]. The presence of kaempferol derivatives in bee pollens is in agreement with existing literature [8,14,16]. Vestigial amounts of kaempferol were also previously detected in honey samples [5,25].

Two myricetin compounds were also identified. Both showed fragment ions at *m/z* 316 and 271, which suggest a myricetin aglycon losses [15]. Myricetin was also found in other bee-pollen products at vestigial amounts (0.04 µg/g of dw) [8,14] and also in honey of *Castanea sativa* [25].

Finally, three forms of isorhamnetin were detected, exhibiting a molecular peak at *m/z* 623 (519 for isorhamnetin acetyl hexoside) and a fragment pattern at *m/z* 315, which is coherent with isorhamnetin aglycon losses [12,13,14]. Their identification in bee pollen was already described in other works [14], as well as in honey [13].

The abundance of flavonols in pollen, namely quercetin derivatives, is in accordance with previous reports, being considered the main reason for the biological potential exhibited by this matrix, including abilities to capture free radicals and reactive species, chelate metals, inhibit cytochrome P450 action, and increase antioxidant defenses [28,29].

### 3.3. Antioxidant Capacity

We decided to perform a general screening of the antioxidative effects of the pollen extract against DPPH^●^. This revealed an ability to reduce the radical in a dose-dependent manner (Table 2 and Figure 2A), exhibiting an IC_50_ value of 695.99 ± 1.69 µg/mL. Even so, the extract was 46 times less efficient than ascorbic acid (IC_50_ = 15.18 ± 0.47 µg/mL).

Additionally, we also tested its capacity against ^●^NO and O_2_^●−^. Although these radicals are products of normal cellular metabolism, their combination promotes the generation of more radicals, including peroxynitrite, which in turn oxidizes low-density lipoprotein and causes irreversible cell damage [1]. The extract also showed potential to scavenge ^●^NO and O_2_^●−^ species in a dose-dependent manner (IC_50_ scores of 1115.28 ± 5.32 and 449.04 ± 2.01 µg/mL, respectively) (Table 2 and Figure 2B,C). Even so, the capturing abilities presented by the extract were lower than those found for the ascorbic acid control (IC_50_ values of 62.77 ± 0.42 and 28.13 ± 0.47 µg/mL for ^●^NO and O_2_^●−^, respectively).

As far as we know, this is the first report concerning the pollen potential to scavenge ^●^NO and O_2_^●−^ species. Regarding DPPH^●^ potential, pollen collected on Parque Nacional da Peneda of Gerês (Portugal) revealed more effectiveness in scavenging DPPH^●^ (IC_50_ = 5.87 µg/mL) [30], as well as Thai and Indian pollens (IC_50_ values of 428.6 and 6.09 µg/mL, respectively) [4,31]. On the other hand, Brazilian pollen showed weaker activity (IC_50_ = 860 µg/mL, respectively) [32]. Besides, Abrantes pollen showed better capturing potential than honey bees (IC_50_ values ranging from 7.5 and 109.0 mg/mL, depending on the country) [33]. The obtained differences are mainly due to different local origins, which influences phenolic levels and consequent biological potential [2].

In fact, the antioxidant activities exhibited by pollen are mainly due to its richness in phenolics, which have well-known abilities due to their ease of transferring hydrogen atoms or electrons to free radicals and reactive species, neutralizing them [1]. This health property is essentially due to their structure, predominantly due to the catechol group on the B ring, the 2,3-double bond, and the conjugated group on the C ring [27]. Besides, this activity is enhanced by a number of hydroxyl groups [1]. These facts are supported by the high correlations found using Pearson’s test. Particularly strong correlations were verified in the DPPH^●^ experiment, caffeoyl hexose (r = 0.8217, *p* < 0.05, *n* = 1), quercetin 7-glucoside-3-*O*-rutinoside (r = 0.9994, *p* < 0.05, *n* = 1), quercetin derivative 1 (r = 0.8869, *p* < 0.05, *n* = 1), quercetin acetyl rhamnoside (r = 0.8092, *p* < 0.05, *n* = 1), and also with total phenolic compounds and total flavonoids (r > 0.7560, *p* < 0.05, *n* = 1). On the other hand, regarding ^●^NO scavenging potential, high correlations were found between this and caffeoyl hexose (r = 0.8769, *p* < 0.05, *n* = 1), quercetin 7-glucoside-3-*O*-rutinoside (r = 0.9620, *p* < 0.05, *n* = 1), quercetin derivative 1 (r = 0.7960, *p* < 0.05, *n* = 1), quercetin acetyl rhamnoside (r = 0.8334, *p* < 0.05, *n* = 1), and total phenolic compounds (r = 0.7581, *p* < 0.05, *n* = 1). These results are not surprising, since it was already reported that the CH = CH-COOH group presented by caffeoyl hexose, together with the catechol group on the B ring, the double bound and 4-oxo-group on the C ring, and the several hydroxyl groups found in quercetin derivatives make these compounds powerful natural antioxidants [1,27]. Furthermore, a high correlation was also reported between DPPH^●^ and ^●^NO antioxidant assays (r = 0.9694, *p* < 0.05, *n* = 1). Even so, negative correlations were found between the both experiments; isorhamnetin acetyl hexoside and quercetin hexoside (r > −0.8054, *p* < 0.05, *n* = 1). Concerning O_2_^●−^ scavenging potential, mild correlations were found between this assay, isorhamnetin acetyl hexoside (r = 0.5385, *p* < 0.05, *n* = 1), quercetin 3-*O*-rutinoside (r = 0.4862, *p* < 0.05, *n* = 1), and quercetin hexoside (r = 0.5385, *p* < 0.05, *n* = 1), while negative correlations were verified between this experiment, total phenolics (r = −0.8197, *p* < 0.05, *n* = 1), flavonols (r = −0.7693, *p* < 0.05, *n* = 1), quercetin 7-glucoside-3-*O*-rutinoside (r > −0.9354, *p* < 0.05, *n* = 1), and quercetin derivative 1 (r = −0.9961, *p* < 0.05, *n* = 1), and also with DPPH^●^ (r = −0.9233, *p* < 0.05, *n* = 1) and ^●^NO (r = −0.8330, respectively, *p* < 0.05, *n* = 1) assays. The obtained correlations are in accordance with others works [4,29,32].

Nevertheless, it is also important to consider the presence of other natural antioxidant compounds, such as vitamins, organic acids, and carotenoids not determined in this work, whose presence also increase pollen’s biological potential [8].

### 3.4. Antidiabetic Capacity

The pollen hydroethanolic extract also exhibits the capacity to inhibit *α*-glucosidase activity, an important enzyme involved in carbohydrate digestion. In this study, this activity was dependent on the concentration, revealing an IC_25_ score of 1192.71 ± 8.14 µg/mL (Table 2 and Figure 2D). However, it was ten times less effective than the acarbose control (IC_25_ = 113.81 ± 1.00 µg/mL).

The antidiabetic effects of pollen have already been described by Daudu et al. [34], who conducted a study involving aqueous extracts of pollen and reported IC_50_ values of 4510 and 600 µg/mL regarding *α*-amylase and *α-*glucosidase inhibitory activities, respectively. Comparatively to other bee products, propolis hydroethanolic extracts (75:25, *v/v*) already showed ability to reduce baker’s yeast *α*-glucosidase action, and rat intestinal sucrase and maltase enzymes (IC_50_ values of 7.24, 32.34 and 71.82 µg/mL) [35]. Besides, pure honey also showed antidiabetic properties. Indeed, diabetic rats which were fed with 1 mg/kg of honey over 6 weeks showed lower levels of serum glucose, creatinine, cholesterol, malondialdehyde, triglycerides, aspartate transaminase, and aspartate aminotransferase, and higher levels of insulin as compared to the untreated group [36]. Furthermore, thirty-two type 2 diabetes mellitus patients who ingested 25 g of honey over 4 months had reduced cholesterol and glycated hemoglobin levels [37].

Indeed, phenolics have already been proven to protect pancreatic *β*-cells from oxidative damage, incentive insulin production [36], and interact with carbohydrate enzyme substrates in both competitive and non-competitive ways, slowing down the breakdown of sugars, and therefore reducing glucose levels in the bloodstream [35]. The mentioned effects are enhanced by phenolic levels and also by their structures; namely the presence of hydroxyl and carbonyl groups [12,38]. In pollen’s case, this capacity is strictly related to the presence of quercetin derivatives. Indeed, the notable ability of quercetin in stopping *α*-glucosidase activity was already described, revealing inhibitory effects of 91% at 200 µM [38]. This fact is corroborated by the mild correlation found between quercetin 3-*O*-rutinoside and *α*-glucosidase inhibitory action (r = 0.5362, *p* < 0.05, *n* = 1).

### 3.5. Protective Effects on Human Erythrocytes

In this study, pollen extracts inhibited hemoglobin oxidation induced by AAPH dose-dependently, with an IC_50_ score of 311.50 ± 1.37 µg/mL (Table 2 and Figure 3A). Still, it was not more efficient than the quercetin control (IC_50_ = 2.61 ± 0.15 µg/mL). Furthermore, it also prevented lipid peroxidation and hemolysis in a concentration-dependent manner, exhibiting 25% inhibitory concentration values of 277.03 ± 2.52 and 103.48 ± 2.23 µg/mL, respectively (Table 2 and Figure 3B,C). Once again, this was less than the quercetin control, whose IC_25_ values obtained were 1.00 ± 0.15 and 0.60 ± 0.15 µg/mL for lipid peroxidation and hemolysis, respectively.

Although this is the first study regarding the capacity of pollen to protect human erythrocytes against hemolysis and lipid peroxidation, Barbieri et al. [39] already reported that 50 µg/mL of bee pollen extract can protect these cells against AAPH-induced oxidation. Regarding other bee products, monofloral honeys were able to diminish lipid peroxidation in human erythrocytes by around 70% at concentrations of 80 µg/mL, and also avoided hemolysis, displaying IC_50_ values varying from 24.68 to 62.56 µg/mL [13]. As far as we know, honey did not show any preventive effects in protecting human erythrocytes against hemoglobin oxidation [40].

Indeed, phenolic compounds, mainly due to their hydroxyl groups and liposolubility, can penetrate the cytoplasm of erythrocytes, and hence scavenge free radicals before they can attack erythrocyte membranes, enhancing their integrity and preventing oxidative injury [1,13,39]. Between phenolics, Peyrat-Maillard, Cuvelier, & Berset [41] already reported the protective effects displayed by caffeic acid, quercetin 3-*O*-rutinoside and quercetin against AAPH induced oxidation with values of 40, 41 and 55 µM/min. Moreover, quercetin and quercetin 3-*O*-rutinoside also showed capacity to prevent hemolysis in bovine erythrocytes, revealing IC_50_ scores of 31 and 37 µM [42], and avoided lipid peroxidation at concentrations below 10 µmol/L [43]. Moreover, the ability of phenolic acids, including caffeic acid, to block lipid peroxidation initiated by metmyoglobin has also been reported [44]. These facts are in line with the positive correlations observed involving hemoglobin oxidation, hemolysis, and quercetin 3-*O*-rutinoside (r > 0.7355, *p* < 0.05, *n* = 1), and between lipid peroxidation, caffeoyl hexose (r = 0.7352, *p* < 0.05, *n* = 1), quercetin rutinoside (r = 0.9000, *p* < 0.05, *n* = 1), and quercetin acetyl rhamnoside (r = 0.7557, *p* < 0.05, *n* = 1). Furthermore, notable correlations were also described regarding the O_2_^●−^ scavenging potential of pollen, anti-hemolytic effects (r = 0.9493, *p* < 0.05, *n* = 1), and between hemoglobin oxidation, lipid peroxidation (r = 0.8096, *p* < 0.05, *n* = 1), and hemolysis (r = 0.7091, *p* < 0.05, *n* = 1). Mild correlations were verified among hemoglobin oxidation and O_2_^●−^ assay (r = 0.5387, *p* < 0.05, *n* = 1). Even so, negative correlations were obtained between hemoglobin oxidation and quercetin derivative 1 (r = −0.6037, *p* < 0.05, *n* = 1), and between hemolysis, total phenolics (r = −0.6770, *p* < 0.05, *n* = 1), total flavonols (r = −0.6928, *p* < 0.05, *n* = 1), quercetin 7-glucoside-3-*O*-rutinoside (r = −0.7769, *p* < 0.05, *n* = 1), quercetin derivative 1 (r = −0.9724, *p* < 0.05, *n* = 1), DPPH^●^ (r = −0.7557, *p* < 0.05, *n* = 1), and ^●^NO experiments (r = −0.6354, *p* < 0.05, *n* = 1). Concerning lipid peroxidation, negative correlations were obtained with isorhamnetin acetyl hexoside (r = −0.7596, *p* < 0.05, *n* = 1) and quercetin hexoside (r = −0.7454, *p* < 0.05, *n* = 1). The obtained correlations reinforce previous results which report that the antioxidant ability of phenolics is strongly influenced by the catechol residue, and number and position of hydroxyl groups, and less by the glycosylation pattern [27]. In fact, the increase of hydroxyl groups, together with the presence of the catechol group, improves this activity, facilitating neutralization of free radicals and reactive species [1,38].

### 3.6. Effect of Pollen Extracts in Mitochondrial Activity and Membrane Integrity

As already mentioned before, it was already reported that pollen has antidiabetic properties and is able to prevent lipid peroxidation [13,34,35]. Therefore, the capacity of the pollen extract to interfere with Caco-2 and HepG2 cancer cells growth was also tested. These cell lines were selected since they are considered models for intestinal epithelium, human toxicology, and metabolism studies, respectively [1,29]. Although pollen extract did not show any selective cytotoxicity for the tested cancer cells (i.e., Caco-2 and HepG2 cells), nor with the NHDF normal cell line (cells viability > 90%), its use as an antidiabetic agent is encouraged by these results combined with its ability to interact with *α*-glucosidase activity [17].

Considering other studies, Sousa et al. [28] verified that the pollen fraction enriched in flavonols protects human adenocarcinoma Caco-2 cells against oxidative damage triggered by *tert*-Butyl hydroperoxide. Furthermore, Oyarzún et al. [29] reported that pollen extracts also showed protective effects against oxidative injury induced by AAPH on Hepa1-6 hepatic cells, finding a mild correlation between this potential and quercetin content (r = 0.64).

## 4. Conclusions

Given the current interest regarding natural products, the present study provides new insights about Abrantes multifloral bee pollen. This regional pollen was portrayed here for the first time and showed notable antioxidant capacity, inhibition of α-glucosidase enzyme action, and protection of human erythrocytes against hemoglobin oxidation, lipid peroxidation, and hemolysis. These health-promoting activities are strongly correlated with phenolic content, highlighting the presence of quercetin derivatives. Furthermore, and although non-anticancer effects were observed, the results obtained herein, namely regarding its antidiabetic action, prove that pollen may be used safely as an antidiabetic agent and incorporated in new pharmaceutical drugs, nutraceuticals, and dietary supplements. The obtained data, together with the verified correlations, can be extrapolated to other pollens with similar compositions. Notwithstanding, further investigation, namely in vivo studies, should be conducted to assess the full biological potential of this bee derivative and to reveal a safe dosage.

## Figures and Tables

**Figure 1 foods-10-02804-f001:**
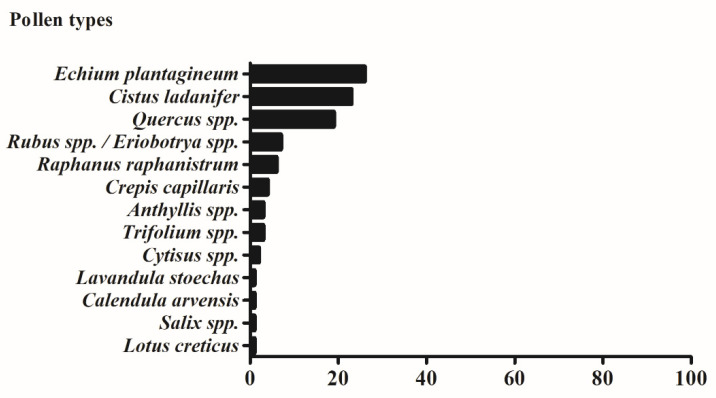
Frequency of the different pollen types in the investigated sample.

**Figure 2 foods-10-02804-f002:**
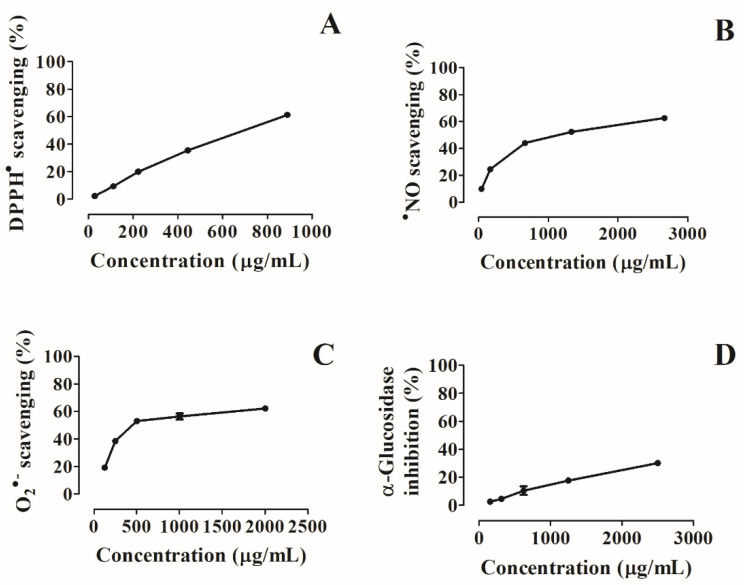
Effects of pollen hydroethanolic extracts against (**A**) 1,1-diphenyl-2-picrylhydrazyl (DPPH^●^), (**B**) nitric oxide (^●^NO), (**C**) superoxide (O2^●−^), and (**D**) *α-*glucosidase inhibition activity.

**Figure 3 foods-10-02804-f003:**
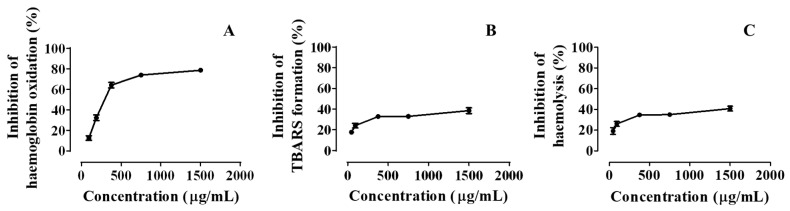
Inhibition of (**A**) hemoglobin oxidation, (**B**) lipid peroxidation, and (**C**) hemolysis of human erythrocytes by extracts of pollen hydroethanolic extracts.

**Table 1 foods-10-02804-t001:** Retention time (Rt), wavelengths of maximum absorption in the visible region (λ_max_) used for quantification, mass fragmentation data, and tentative identification of quantified peaks of compounds (µg/g of dry weight) in pollen.

Peak	Compounds Identification	HPLC-DAD-ESI-MS^n^ Data				Quantification
		R_t_ (min)	λ_max_ (nm)	Molecular Ion [M-H] (*m/z*)	Fragments MS/MS (*m/z*)	
1	Caffeoyl di-hexoside	10.0	320	635	341, 179	nq
2	Coumaroyl hexose	13.8	320	325	145, 163, 205, 235	nq
3	Caffeoyl hexose	14.0	320	341	179, 135	0.056 ± 0.0057
4	Quercetin 7-glucoside-3-*O*-rutinoside	24.0	350	771	609, 301	0.35 ± 0.020
5	Kaempferol di-hexoside	24.3	350	609	447, 285	nq
6	Myricetin rhamno-hexoside	25.0	350	625	316, 271, 287	nq
7	Quercetin 3-*O*-rutinoside	25.9	350	609	271, 301	0.76 ± 0.037
8	Kaempferol 3-*O*-rutinoside-*O*-hexoside	26.2	350	755	593, 447, 285	nq
9	Quercetin derivative 1	26.3	350	639	314, 301, 150	0.49 ± 0.031
10	Myricetin derivative	26.4	350	521	316; 271	nq
11	Isorhamnetin 3-*O*-rutinoside 1	26.8	350	623	315	nq
12	Kaempferol 3-*O*-rutinoside	27.3	350	447	285, 256	nq
13	Quercetin 3-*O*-glucoside	28.4	350	463	300/301, 271	nq
14	Isorhamnetin 3-*O*-rutinoside 2	28.5	350	623	315	nq
15	Quercetin derivative 2	28.6	350	609	315, 300, 271	nq
16	Quercetin acetyl rhamnoside	29.3	350	505	463, 301	1.33 ± 0.022
17	Isorhamnetin acetyl hexoside	31.7	350	519	315	0.82 ± 0.013
18	Kaempferol acetyl hexoside	31.9	350	447	235	nq
19	Kaempferol hexoside	32.6	350	489	447, 285	nq
20	Quercetin hexoside	35.0	350	463	301	0.36 ± 0.032
Σ						4.17

Values are expressed as mean ± standard deviation of three assays. Σ, sum of the determined phenolic compounds; nq, not quantified.

**Table 2 foods-10-02804-t002:** **The 25% inhibitory concentration (IC_25_)** and half maximal inhibitory concentration (IC_50_) (µg/mL) values found in the antioxidant activity, *α-*glucosidase, hemoglobin oxidation, and hemolysis assays of pollen hydroethanolic extracts.

	DPPH^●^	^●^NO	O_2_^●−^	*α*-Glucosidase	Haemoglobin Oxidation	Haemolysis	Lipid Peroxidation
**Extract**	695.99 ± 1.69	1115.28 ± 5.32	449.04 ± 2.01	**1192.71 ± 8.14**	311.50 ± 1.37	**103.48 ± 2.23**	**277.03 ± 2.52**

Values are expressed as mean ± standard deviation of three assays concerning the antioxidant capacity against 1,1-diphenyl-2-picrylhydrazyl, nitric oxide and superoxide radicals (DPPH^●^, ^●^NO and O_2_^●−^, respectively), *a*-glucosidase inhibitory activity, hemoglobin oxidation, hemolysis and lipid peroxidation. **IC_25_: 25% inhibitory concentration**; IC_50_: 50% inhibitory concentration.

## Data Availability

Data are contained within this article.

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
