# Peer review of "Potential Activity of Abrantes Pollen Extract: Biochemical and Cellular Model Studies"

_foods, 2021, doi:10.3390/foods10112804_

Round 1
Reviewer 1 Report
This study is a report of basic studies on the polyphenol component and antioxidant capacity in Portuguese pollen extract. However, no new reports are found. In addition, it seems necessary to reconsider the points described below.
(1) Many markers have been examined, but it is necessary to state that these markers make little sense in an actual animal and human.
(2) Pollen data was one-year, and it is possible to predict that the composition of pollen extract was changed and the markers will be changed due to the environment such as climatic conditions, so it is necessary to describe it.
(3) Authors evaluated the safety using culture cells. However, these data were just cytotoxicity and was not called a safety evaluation.
(4) Inhibition of enzymes such as glucosidase is an overstatement of anti-diabetic effect.
(5) P2, L51: 'flavanol' was changed 'flavonol'.
Author Response
This study is a report of basic studies on the polyphenol component and antioxidant capacity in Portuguese pollen extract. However, no new reports are found. In addition, it seems necessary to reconsider the points described below.
Authors’ response: First of all, we would like to thank your kind comments and compliments made to our manuscript. Following the comments received, the changes made by us are highlighted in the revised version, in accordance with the request. Even so, after each Reviewer’s comment, we indicated the main changes introduced and the corresponding lines.
(1) Many markers have been examined, but it is necessary to state that these markers make little sense in an actual animal and human.
Authors’ response: Thank you so much for your commentary. As we know, nowadays, several evidence already reported that pollen, namely their content in phenolic compounds, can be undoubtedly considered a floral marker, since determined phenolics are only /or mostly found in specific honeys. Therefore, it is not surprising that pollen is considered an authenticity maker of different types of honey. Please see, for example:
- Andrade et al. (1997), Determination of phenolic compounds in honeys with different floral origin by capillary zone electrophoresis, 60(1), 79*84. https://doi.org/10.1016/S0308-8146(96)00313-5
- Ferreres et al. (1995), Floral nectar phenolics as biochemical markers for the botanical origin of heather honey. European Food Research and Technology 202(1), 40-44. https://doi.org/0.1007/BF01229682
- Silva et al. (2021), Chemical and Antioxidant Characterization of the Portuguese Heather Honey from Calluna vulgaris. Separations, 8, 177. https://doi.org/10.3390/separations8100177
- Truchado et al. (2008), Nectar Flavonol Rhamnosides Are Floral Markers of Acacia (Robinia pseudacacia) Honey, Journal of Agricultural and Food Chemistry, 56 (19), 8815–8824. https://doi.org/10.1021/jf801625t
- Tomás-Barberán, F.A. et al. (2001), HPLC flavonoid profiles as markers for the botanical origin of European unifloral honeys. Journal of the Science of Food and Agriculture, 81: 485-496. https://doi.org/10.1002/jsfa.836
- Ferreres et al. (1996), Natural Occurrence of Abscisic Acid in Heather Honey and Floral Nectar. Journal of Agricultural and Food Chemistry, 44, 2053-2056. https://10.1021/JF9507553
- Ferreres et al. (1994), Flavonoids from Portuguese heather honey. Z Lebensm Unters Forch ,199, 32–37. https://doi.org/10.1007/BF01192949
- Silva et al. (2020), Authentication of honeys from Caramulo region (Portugal): Pollen spectrum, physicochemical characteristics, mineral content, and phenolic profile. Journal of Food Science, 85: 374-385. https://doi.org/10.1111/1750-3841.15023
- Escuredo et al. (2012), Assessing Rubus honey value: Pollen and phenolic compounds content and antibacterial capacity. Food Chemistry, 130(3), 671-678. https://10.1016/j.foodchem.2011.07.107
(2) Pollen data was one-year, and it is possible to predict that the composition of pollen extract was changed and the markers will be changed due to the environment such as climatic conditions, so it is necessary to describe it.
Authors’ response: Thank you so much for your commentary. You have totally reason. In fact, pollen composition depends on several factors, e.g., meteorological conditions, storage time and type, among others. In order to complete the article, this information was added (please see now lines 967 to 971, and 981 to 985 of the revised version).
(3) Authors evaluated the safety using culture cells. However, these data were just cytotoxicity and was not called a safety evaluation.
Authors’ response: First of all, thank you so much for your commentary. In fact, we used three different cellular lines, two are cancer cells (human adenocarcinoma Caco-2 and human liver HepG2 cells) and another one is normal cells, i.e., fibroblasts (NHDF). First of all, we decided to understand the cytotoxic tests regarding the effects of pollen on cancer cells, and although we didn’t see notorious effects of this extract on this type of cells, the obtained results together with its ability to interact with α-glucosidase activity encourage the use of pollen as antidiabetic agent. Additionally, and in order to also deepen the knowledge about the cytotoxic selectivity of pollen, we also performed the same assays in NHDF cells. The extract showed also to be non-cytotoxic in these normal cells. We apologize but, at the moment it is impossible for us to perform these assays with normal colon and liver cells (please see lines 1272 to 1287 of the revised version). Even so, the use of NHDF cells for comparative purposes is routinely used. Please see, for example:
- Giusti et al. (2018). Ovarian cancer-derived extracellular vesicles affect normal human fibroblast behavior. Cancer cell and therapy, 19(8), 722–734. https://10.1080/15384047.2018.1451286
- Gonçalves et al. (2020), Multitarget protection of Pterospartum tridentatum phenolic-rich extracts against a wide range of free radical species, antidiabetic activity and effects on human colon carcinoma (Caco-2) cells. Journal of Food Sciences. https://10.1111/1750-3841.15511
- Wessels et al. (2019). Reciprocal signaling and direct physical interactions between fibroblasts and breast cancer cells in a 3D environment, PLOS ONE, 14(6), e0218854. https://10.1371/journal.pone.0218854
- Lima et al. (2020), Photodynamic activity of indolenine-based aminosquaraine cyanine dyes: Synthesis and in vitro photobiological evaluation. Dyes and Pigments, 174, 108024. https://doi.org/10.1016/j.dyepig.2019.108024
(4) Inhibition of enzymes such as glucosidase is an overstatement of anti-diabetic effect.
Authors’ response: Thank you so much for your commentary. In fact, the inhibition of enzymes, such as α-glucosidase, which is responsible for breakdown starch and disaccharides to glucose, are considered screening assays, and allow us to make a first evaluation about the antidiabetic capacities of extracts and/or natural products in order to understand if it will be valuable to deepen results in cells, and after in animals and humans. Assays similar to that were largely found in other articles, for example:
- Jesus et al. (2018), Exploring the phenolic profile, antioxidant, antidiabetic and anti-hemolytic potential of Prunus avium vegetal parts. Food Research International. https://10.1016/j.foodres.2018.08.079
- Tadera et al. (2006), Inhibition of alpha-glucosidase and alpha-amylase by flavonoids. Journal of Nutritional Science and Vitaminology, 52(2), 149-153. https://10.3177/jnsv.52.149.
- Gonçalves et al. (2020), Multitarget protection of Pterospartum tridentatum phenolic-rich extracts against a wide range of free radical species, antidiabetic activity and effects on human colon carcinoma (Caco-2) cells. Journal of Food Sciences. https://10.1111/1750-3841.15511
- Umar et al. (2019). Untargeted metabolomics analysis using FTIR and UHPLC-Q-Orbitrap HRMS of two Curculigo species and evaluation of their antioxidant and α-glucosidase inhibitory activities. Metabolites, 11, 42. https://10.3390/metabo11010042
- Bento et al. (2018), Assessing the phenolic profile, antioxidant, antidiabetic and protective effects against oxidative damage in human erythrocytes of peaches from Fundão. Journal of Functional Foods, 43. https://10.1016/j.jff.2018.02.018
- Daudu et al. (2019), Bee pollen extracts as potential antioxidants and inhibitors of α-amylase and α-glucosidase enzymes - In vitro assessment. Journal of Apicultural Science, 63(2), 315-325. https://10.2478/jas-2019-0020
- Silva et al. (2015). Phenolic profile and biological potential of Endopleura uchi extracts. Asian Pacific Journal of Tropical Medicine, 8(11), 889-897. https://10.1016/j.apjtm.2015.10.013
- Gonçalves et al. (2018), Antioxidant status, antidiabetic properties and effects on Caco-2 cells of colored and non-colored enriched extracts of sweet cherry fruits. Nutrients, 10(11), 1688. https://10.3390/nu10111688
- Zhang et al. (2015), Inhibitory properties of aqueous ethanol extracts of propolis on alpha-glucosidase. Evidence-Based Complementary and Alternative Medicine. https://10.1155/2015/587383
(5) P2, L51: 'flavanol' was changed 'flavonol'.
Authors’ response: Thank you so much for your note, the same as corrected (please see now line 69 of the revised version).
Reviewer 2 Report
Some minor modifications are required.
correct the Celcius symbol in line 229 and 234
Please delete the unnecessary preamble from the Results and Discussion
Line 288-294 (The study of the hive surrounding flora, .......... shelf-life [2].)
Line 319-322 (Phenolics presence contributes........... humidity, and soil type [7].)
Line 385-396 (It is recognized the role of ........ health promotor [5]. Therefore,)
Line 464-466: Diabetes mellitus is .......or fewer side effects [1].
Line 495-499: Erythrocytes are the most abundant ..........bee products [13,39].
Figure 3 is not legible. Please enlarge the figures

Author Response
Some minor modifications are required.
correct the Celcius symbol in line 229 and 234
Authors’ response: Thank you so much for your note, the same as corrected (please see now lines 304, 314, 587, 599 and 614 of the revised version).
Please delete the unnecessary preamble from the Results and Discussion
Authors’ response: Thank you so much for your commentary, the same was taken into account and the unnecessary preambles eliminated (please see the revised version).
Line 288-294 (The study of the hive surrounding flora, .......... shelf-life [2].)~
Authors’ response: Thank you so much for your commentary, the same as deleted (please see the revised version).
Line 319-322 (Phenolics presence contributes........... humidity, and soil type [7].)
Authors’ response: Thank you so much for your commentary, the same as deleted (please see revised version).
Line 385-396 (It is recognized the role of ........ health promotor [5]. Therefore,)
Authors’ response: Thank you so much for your commentary, the same as deleted (please see the revised version).
Line 464-466: Diabetes mellitus is .......or fewer side effects [1].
Authors’ response: Thank you so much for your note, the same as deleted (please see the revised version).
Line 495-499: Erythrocytes are the most abundant ..........bee products [13,39].
Authors’ response: Thank you so much for your commentary, the same as deleted (please see the revised version).
Figure 3 is not legible. Please enlarge the figures
Authors’ response: Thank you so much for your suggestion, the figure was enlarged (please see now figure 3 of the revised version).
Reviewer 3 Report
The work "Potential Activity of Abrantes Pollen Extract: Biochemical and Cellular Model Studies" tries to highlight the properties of the pollen from Abrantes by evaluating its antioxidative and antidiabetic properties.
After reading it, I indicate below my doubts and questions to the authors:
Abbreviations in the abstract should be avoid or defined
Line 96 and 200 : Change rpm for its equivalent in g
Line 100: 45×, what does it mean?
Change the sentence " Injections were performed in triplicate".
Section 2.7: Explain in more detail and give some references where this procedure has been used previously.
In the sentences:
"Three experiments were performed in triplicate" and "Seven different concentrations were prepared"
Their use sounds reiterative.
Also, the range of concentrations used is not stated sometimes but included others (i.e. line 196). Please try to correct this point
Section 2.9.2 should go before 2.9.1.
Section 2.9.2: with respect to the patients' blood samples. Before their inclusion, Were they checked for any diseases and/or other characteristics that could interfere with the parameters to be determined?
Line 235: Please, do not start a sentence with a number written in digits.
I do not understand why quecerten was used as a positive control, even in processes that measure processes such as hemolysis or inhibition. Indicate any previous work or justification as to why this substance is eligible as a positive control.
It is not very clear to this reviewer why the choice of cell lines used was made. The use of erythrocytes should also be further justified.
Explain why do you selected the cells models used in this work.
2.10.3. Membrane integrity assay: indicate any references that have used this method.
308-314: this information would be better placed in the introduction.
Figure 3: too small, no indication of the number of experiments, no definition of the abbreviations used.
Personally, I do not like that discussion and results go together. It is more difficult to follow and also makes the conclusion section look like a hybrid conclusion-discussion, too short to be a discussion, but too long and convoluted to be a conclusion. In my opinion, conclusions should be more direct and summarized.
The presentation of the paper should be improved, especially the figures.
In material and methods, a short introductory sentence on the justification of the technique used would help the reader to understand and follow the reasoning that has led the authors to propose such experiments.
Author Response
The work "Potential Activity of Abrantes Pollen Extract: Biochemical and Cellular Model Studies" tries to highlight the properties of the pollen from Abrantes by evaluating its antioxidative and antidiabetic properties.
After reading it, I indicate below my doubts and questions to the authors.
Authors’ response: First of all, we would like to thank your kind comments and compliments made to our manuscript. Following the comments received, the changes made by us are highlighted in the revised version, in accordance with the request. Even so, after each Reviewer’s comment, we indicated the main changes introduced and the corresponding lines.
Abbreviations in the abstract should be avoid or defined
Authors’ response: Thank you so much for your note, the same were defined (please see now lines from 23-24 and 26 of the revised version).
Line 96 and 200 : Change rpm for its equivalent in g
Authors’ response: Thank you so much for your note, but unfortunately, the centrifuge used is broken and it is impossible for us to discover the rotor radius, and therefore, it is impossible for us to change rpm to g, sorry.
Line 100: 45×, what does it mean?
Authors’ response: Thank you so much for your note. 45x was the used microscopic lens; the sentence was clarified in the article (please see now lines 159 and 160 of the revised version).
Change the sentence " Injections were performed in triplicate".
Authors’ response: Thank you so much for your suggestion. The sentence was changed (please see now lines 193 of the revised version).
Section 2.7: Explain in more detail and give some references where this procedure has been used previously.
Authors’ response: Thank you so much for your note. In fact, section 2.7 was only a brief explanation regarding the assays done; them were explained below (please see sub-sections 2.7.1., 2.7.2., and 2.7.3. of the revised version).
In the sentences:
"Three experiments were performed in triplicate" and "Seven different concentrations were prepared". Their use sounds reiterative. Also, the range of concentrations used is not stated sometimes but included others (i.e. line 196). Please try to correct this point
Authors’ response: Thank you so much for your commentary. The same was taken into account and all was standardized. We decided to remove the tested concentrations on materials and methods section to avoid repetitions once they are mentioned in discussion, figures and tables (please see now lines 201 to 202 and subsections 2.7.1,2.7.2 and 2.7.3).
Section 2.9.2 should go before 2.9.1.
Authors’ response: Thank you so much for your interesting commentary, the same was tacked into account (please see now lines from 309 and 320 of the revised version).
Section 2.9.2: with respect to the patients' blood samples. Before their inclusion, Were they checked for any diseases and/or other characteristics that could interfere with the parameters to be determined?
Authors’ response: Thank you very much for your pertinent comment. As mentioned (please see now line 310 of the revised version), venous human blood samples were collected from randomized patients (this is the only way that the hospital lets us obtain samples to perform this type of assays). Therefore, it is not necessary to add the proper ethical clearance and check for any diseases and/or other characteristics.
Line 235: Please, do not start a sentence with a number written in digits.
Authors’ response: Thank you very much for your pertinent note, the same was corrected (please see now lines 614 and 615 of the revised version
I do not understand why quecerten was used as a positive control, even in processes that measure processes such as hemolysis or inhibition. Indicate any previous work or justification as to why this substance is eligible as a positive control.
Authors’ response: Thank you very much for your pertinent comment. Quercetin is routinely used as control positive in this type of assay given its notable potential to protect human erythrocytes from damage. These health benefits are due to its remarkable antioxidant properties, which, in turn, are closely related to its many hydroxyl groups. Please, see:
- Asgary, S., Naderi, G. H., & Askari, N. (2005). Protective effect of flavonoids against red blood cell hemolysis by free radicals. Experimental and Clinical Cardiology, 10(2), 88–90
- - Balaji, B., Rajendar, B., & Ramanathan, M. (2014). Quercetin protected isolated human erythrocytes against mancozeb-induced oxidative stress. Toxicology and Industrial Health, 30(6), 561–569. https://doi.org/10.1177/0748233712462465
- Chisté, R. C., Freitas, M., Mercadante, A. Z., & Fernandes, E. (2014a). Carotenoids are effective inhibitors of in vitro hemolysis of human erythrocytes, as determined by a practical and optimized cellular antioxidant assay. Journal of Food Science, 79(9), H1841–H1847. https://doi.org/10.1111/1750-3841.12580
- Chisté, R. C., Freitas, M., Mercadante, A. Z., & Fernandes, E. (2014b). Carotenoids inhibit lipid peroxidation and hemoglobin oxidation, but not the depletion of glutathione induced by ROS in human erythrocytes. Life Sciences, 99(1–2), 52–60. https://doi.org/10.1016/j.lfs.2014.01.059
- Ferrali, M., Signorini, C., Caciotti, B., Sugherini, L., Ciccoli, L., Giachetti, D., & Comporti, M. (1997). Protection against oxidative damage of erythrocyte membrane by the flavonoid quercetin and its relation to iron chelating activity. FEBS Letters, 416(2), 123–129. https://doi.org/10.1016/S0014-5793(97)01182-4
- Gonçalves, Ana C, Rodrigues, M., Santos, A. O., Alves, G., & Silva, L. R. (2018). Antioxidant status, antidiabetic properties and effects on Caco-2 cells of colored and non-colored enriched extracts of sweet cherry fruits. Nutrients, 10(11), 1688. https://doi.org/10.3390/nu10111688
- Gonçalves, Ana Carolina, Bento, C., Nunes, A. R., Simões, M., Alves, G., & Silva, L. R. (2020). Multitarget protection of Pterospartum tridentatum phenolic-rich extracts against a wide range of free radical species, antidiabetic activity and effects on human colon carcinoma (Caco-2) cells. Journal of Food Science, 85(12), 4377–4388. https://doi.org/10.1111/1750-3841.15511
- Jesus, F., Gonçalves, A. C., Alves, G., & Silva, L. R. (2018). Exploring the phenolic profile, antioxidant, antidiabetic and anti-hemolytic potential of Prunus avium vegetal parts. Food Research International. https://doi.org/10.1016/j.foodres.2018.08.079
- Mikstacka, R., Rimando, A. M., & Ignatowicz, E. (2010). Antioxidant effect of trans-Resveratrol, pterostilbene, quercetin and their combinations in human erythrocytes In vitro. Plant Foods for Human Nutrition, 65(1), 57–63. https://doi.org/10.1007/s11130-010-0154-8
It is not very clear to this reviewer why the choice of cell lines used was made. The use of erythrocytes should also be further justified.
Explain why do you selected the cells models used in this work.
Authors’ response: Thank you very much for your commentary. The explanation was added (please see now lines 93 to 98 of the revised version).
2.10.3. Membrane integrity assay: indicate any references that have used this method.
Authors’ response: Thank you very much for your note, the reference was used (please see now line 902 of the revised version).
308-314: this information would be better placed in the introduction.
Authors’ response: Thank you so much for your suggestion. This part of the discussion was changed (please see now the revised version).
Figure 3: too small, no indication of the number of experiments, no definition of the abbreviations used.
Authors’ response: Thank you so much for your suggestion, the figure was enlarged (please see now figure 3 of the revised version).
Personally, I do not like that discussion and results go together. It is more difficult to follow and also makes the conclusion section look like a hybrid conclusion-discussion, too short to be a discussion, but too long and convoluted to be a conclusion. In my opinion, conclusions should be more direct and summarized.
Authors’ response: Thank you so much for your pertinent commentary. Even so, all authors discussed and conclude that, since the article present many and different results, it will be better for the readers to read results and discussion together to, in part, clarify all information and avoid having to go backward. We hope you understand and agree with us.
The presentation of the paper should be improved, especially the figures.
Authors’ response: Thank you so much for your suggestion, the figures were all improved (please see now the revised version).
In material and methods, a short introductory sentence on the justification of the technique used would help the reader to understand and follow the reasoning that has led the authors to propose such experiments.
Authors’ response: Thank you so much for your note, the same was added (please see now lines 155, 173, 180, 196-197, 205, 280, 291, 301, 322-323, 605-607, 619 and 633).
Round 2
Reviewer 1 Report
I think the author is doing enough, so I accept it.
Reviewer 3 Report
“Authors’ response: Thank you very much for your pertinent comment. As mentioned (please see now line 310 of the revised version), venous human blood samples were collected from randomized patients (this is the only way that the hospital lets us obtain samples to perform this type of assays). Therefore, it is not necessary to add the proper ethical clearance and check for any diseases and/or other characteristics”
Comment to author´s response: In such a case, this point should be included in the discussion indicating that it is a limitation.